# Transmission, localization, and infectivity of seedborne maize chlorotic mottle virus

**Pauline Bernardo[1], Kelly Barriball[2], Timothy S. Frey[1], Tea Meulia[3], Anne Wangai[4], L. M. Suresh[5], Scott Heuchelin[6], Pierce A. Paul[1], Margaret G. Redinbaugh[1,2]\*, Erik W. Ohlson[2]\***

**1** Department of Plant Pathology, The Ohio State University, Wooster, OH, United States of America, **2** Corn, Soybean, and Wheat Quality Research Unit, United States Department of Agriculture–Agricultural Research Service (USDA-ARS), Wooster, OH, United States of America, **3** Molecular and Cellular Imaging Center, Ohio Agricultural Research and Development Center, Wooster, OH, United States of America, **4** Kenya Agricultural and Livestock Research Organization (KALRO), NARL, Nairobi, Kenya, **5** International Maize and Wheat Improvement Center (CIMMYT), ICRAF, Gigiri, Nairobi, Kenya, **6** Corteva Agriscience, Johnston, IA, United States of America

\* redinbaugh.2@osu.edu (MGR); erik.ohlson@usda.gov (EWO)

**Data Availability Statement:** All relevant data are within the paper and its Supporting Information files.

**Funding:** This work was funded by American Seed Research Foundation project "Seed transmission

## Abstract

Maize lethal necrosis is a destructive virus disease of maize caused by maize chlorotic mottle virus (MCMV) in combination with a virus in the family *Potyviridae*. Emergence of MLN is typically associated with the introduction of MCMV or its vectors and understanding its spread through seed is critical for disease management. Previous studies suggest that although MCMV is detected on seed, the seed transmission rate of this virus is low. However, mechanisms influencing its transmission are poorly understood. Elucidating these mechanisms is crucial for informing strategies to prevent spread on contaminated seed. In this study, we evaluated the rate of MCMV seed transmission using seed collected from plants that were artificially inoculated with MCMV isolates from Hawaii and Kenya. Grow-out tests indicated that MCMV transmission through seed was rare, with a rate of 0.004% among the more than 85,000 seed evaluated, despite detection of MCMV at high levels in the seed lots. To understand factors that limit transmission from seed, MCMV distribution in seed tissues was examined using serology and immunolocalization. The virus was present at high levels in maternal tissues, the pericarp and pedicel, but absent from filial endosperm and embryo seed tissues. The ability to transmit MCMV from seed to uninfected plants was tested to evaluate virus viability. Transmission was negatively associated with both seed maturity and moisture content. Transmission of MCMV from infested seed dried to less than 15% moisture was not detected, suggesting proper handling could be important for minimizing spread of MCMV through seed.

## Introduction

Maize lethal necrosis (MLN) is a virus disease of maize caused by co-infection by *maize chlorotic mottle virus* (MCMV, genus *Machlomovirus;* family *Tombusviridae*) and a virus in the family *Potyviridae*. First discovered in the 1970s in Peru, MCMV has since been reported in North America, Southeast Asia, East Africa, and Western Europe [1–8]. The emergence of

of maize chlorotic mottle virus" and the Bill and Melinda Gates Foundation Project (OPP1138693). Salary and research support for P.A. Paul were provided by state and federal funds to the Ohio Agricultural Research and Development Center. The funders had no role in study design, data collection and analysis, decision to publish, or preparation of the manuscript.

**Competing interests:** The authors have declared that no competing interests exist.

MLN in East Africa in the 2010s has cost smallholder farmers up to $339 million USD annually [9]. The introduction of MCMV in Southeast Asia has also had devastating effects, resulting in yield losses of $2 billion USD in China and substantial losses elsewhere in East Asia [10, 11]. Given the ubiquity of maize infecting potyviruses, most notably sugarcane mosaic virus (SCMV), emergence of MLN is typically associated with introduction of MCMV and the presence of one or more of its vectors [12]. Thus, understanding the spread of MCMV through seed and other means is critical for developing effective MLN management strategies.

Like most tombusviruses, MCMV is easily mechanically transmitted and can be transmitted by insect vectors and through soil [13–19]. MCMV is also transmitted vertically through seed [20], albeit at very low frequency. Nonetheless, seed transmission has likely played a role in the intercontinental spread of MCMV [12]. To date, few studies have evaluated MCMV transmission through seed, the majority of which have reported null to low transmission rates, ranging from 0% to 0.6% [20–24]. Seed transmission rates as high as 8–12% have been reported; however, none of the MCMV positive plants were symptomatic in this study [7]. In other systems, seed transmission rates as low as 0.1% were demonstrated to be sufficient to initiate virus epidemics in crops in the presence of vector populations capable of sustaining secondary transmission [25, 26].

To prevent virus transmission from occurring through seed, it is important to understand the mechanisms through which transmission occurs and the localization of virus within seed. However, these processes are currently poorly understood. While MCMV is present and detectable on seed from infected plants [27], little is known about the viability and localization of the virus in seed. The localization of viruses within plant seed tissues is an important indicator of transmissibility and can inform disease management. Embryo infection has been shown to facilitate disease transmission, allowing several plant viruses to retain viability and transmissibility to the seedling [28–30]. However, there are known exceptions, such as tobamoviruses, where seed transmissibility is high despite uninfected embryos, possibly due to the genus's high stability [31–33]. Furthermore, if the virus is only peripherally present on the seed surface, the virus may have low viability and/or seed treatments may be effective [34–36]. Thus, elucidating localization of MCMV in the seed is an important predictor of seed transmission.

In this study, we investigate the seedborne transmission of MCMV in seed obtained from plants artificially inoculated with MCMV strains from Hawaii and Kenya. Localization of MCMV in and on seed acquired from artificially inoculated maize and purchased in Kenyan markets was evaluated. Transmission and localization were assayed by double antibody sandwich–enzyme linked immunosorbent assay (DAS-ELISA) or immunofluorescence. Infectivity of MCMV obtained from virus infected seed was examined in seed lots from Hawaii and Kenya, and at three seed maturity and moisture content levels by vascular puncture inoculation (VPI).

## Materials and methods

### Seed material

MCMV susceptible maize (*Zea mays* L.) inbred line Oh28 is maintained by USDA-ARS in Wooster, OH. The sweet corn hybrid, Early Sunglow (Park Seed Company, Greenwood, SC, USA), was acquired commercially. MCMV infested maize seed was obtained from Hawaii (MCMV-HI) and Kenya (MCMV-KE). Maize inbred lines CML333 and CML545, developed and originally provided by the International Maize and Wheat Improvement Center (CIMMYT), were grown in Ohio to provide seed for planting in Waialua, HI. Plants were inoculated with MCMV-HI using a mist-blower, as previously described, which can inoculate mechanically transmitted viruses at up to >99% infection efficiency [37, 38]. Similarly, seed from

MCMV-KE infected inbred lines CML333, CML442, and CML545 were collected from artificially inoculated maize lines grown in Njoro, Kenya and provided by the Kenya Agricultural and Livestock Research Organization (KALRO; Nairobi, Kenya).

To assess the impact of maturity and seed moisture content on MCMV infectivity, MCMV-HI infected CML333 and CML545 ears were collected at maize reproductive growth stages R2 (blister), R4 (dough), and R6 (physiological maturity) [39] and shipped to Wooster. MCMV infested seed were imported from Hawaii and Kenya to Wooster, OH under APHIS PPQ permits P526P-15-04764 and P526P-16-00404, respectively. All seed were stored at 4˚C in sealed bags, except for those collected at R2 and R4, which were sealed and stored at -20˚C. MCMV infested and un-infested seed lots used for localization and transmission experiments were described previously [27].

## Transmission of MCMV through seed

Transmissibility of MCMV through seed was evaluated by grow-out tests of CIMMYT maize line seed from Hawaii and Kenya under growth chamber conditions followed by DAS-ELISA. Individual kernels were planted in 288 cell plug trays filled with greenhouse soil mix containing 0.0014% (w/w) Marathon to eliminate potential vectors (OHP, Inc., Bluffton, SC, USA). Plug trays were placed on top of 1020 trays filled with the same soil. Trays were moved to a growth chamber with a 12 h light/12 h dark cycle at 25/21˚C, respectively. Light intensity was maintained at approximately 26,000 $l_v$. Maize seedlings (Early Sunglow) grown in 7.6 cm pots under the same conditions served as healthy controls. Maize inbred Oh28 seedlings inoculated with the Kansas isolate of MCMV (MCMV-KS) served as positive controls [40]. Insect vector presence in the growth chamber was monitored using yellow sticky cards and no vectors were detected throughout the course of the experiments.

Seed emergence rates were determined 14 days after planting (DAP) by counting the number of visible seedlings. At 21 DAP, two leaf discs from each plant in a twelve-cell column were collected using a 6.4-mm-diameter tissue punch and pooled (MIDCO Global, Kirkwood, MO, USA). Preliminary experiments indicated that the pooling strategy was sufficiently sensitive to detect the presence of a single infected leaf disc among >35 uninfected discs (S1 Table). If individual plants were too small for sampling by leaf discs, a section of leaf approximately the size of two tissue discs was collected and added to the pool. The punch was thoroughly cleaned with 70% bleach between each pool. Above soil portions of deceased plants and those too small for tissue collection were collected as previously described and pooled separately; hereafter referred to as 'unhealthy plant' pools. Leaf discs were collected from healthy and positive control plants separately. Samples were stored on ice throughout collection.

Leaf discs were ground in 100 µl General Extraction Buffer (GEB; Agdia, Inc.) using a KLECO model 2-96-A tissue pulverizer (Visalia, CA, USA). Following grinding, 400 µl GEB were added to each sample. Pools of poorly growing or deceased plant samples were weighed and extracted in 2 volumes (w/v) GEB. For each sample, two 100 µl aliquots were tested for the presence of MCMV by DAS-ELISA (Agdia, Elkhart, IN, USA) as previously described [27]. Within each plate, MCMV positive controls were separated from other samples by a row of buffer to prevent and monitor possible cross-contamination. Samples were considered positive if the mean absorbance was greater than twice the mean absorbance of the healthy controls (2HC). To ensure positive pools resulted from seed transmission of MCMV and not from cross-contamination, the individual plants from which discs in positive pools were collected were resampled as described above. RNA was extracted from each individual and used for RT-PCR as previously described [27, 41]. To minimize the presence of false negative tests, samples from pools with mean absorbances greater than the mean of the healthy controls plus

three times the standard deviation of the mean absorbance of the healthy controls (>HC + 3SD) were resampled and retested by ELISA or RT-PCR.

## Localization of MCMV in maize seed tissues

Previously described MCMV positive (KE-G) and negative (HI-A) commercial seed lots [27] were used to evaluate the localization of the virus among maize seed tissues. Seeds were incubated overnight at 30°C in distilled water, then treated in one of three ways: 1) dissected by hand directly into pericarp, pedicel, endosperm, and embryo tissues using forceps, 2) washed with a 15% bleach, 0.01% Triton-X-100 solution for 15 min on a shaker prior to hand dissection, or 3) washed with the bleach/Triton solution, followed by excision of the embryo by hand dissection and then washing of the excised embryos for 5 min on a shaker with the bleach/Triton solution. Forceps were sterilized in a Bacti-Cinerator™ IV Sterilizer between dissection of each tissue and each seed (Technical Products International, Inc., St. Louis, MO, USA) to minimize transfer of MCMV between tissues and samples. Samples consisted of individual tissues from each treated seed/embryo homogenized by pulverization in GEB (1:6 w/v), then evaluated for the presence of MCMV by DAS-ELISA.

## Immunocytochemical localization of MCMV in maize seed

MCMV-positive (HI-H, HI-F, KE-D, KE-F and KE-G) and MCMV-free seed (HI-A, HI-B, KE-C) were incubated at 30°C in sterile water overnight. Longitudinal sections (ca. 2 mm) through the plane of the scutellum and embryo or cross-sections through the radicle and plumule, were collected and fixed in 4% paraformaldehyde at room temperature for 3 h. Samples were then dehydrated and wax embedded by: 4% paraformaldehyde (2x 45 min); 80% ethanol (70 min); 90% ethanol (80 min), 95% ethanol (60 min), 100% ethanol (2x 70 min), Pro-Par (50 min; Anatech Ltd, MI, USA); and paraffin (2x 50 min; 63°C; Polyscience, Niles, Il, USA; cat #19562) baths. The 10 μm paraffin embedded sections were collected onto silane coated slides (Polyscience) using a RM2265 microtome (Leica Biosystems, Wetzlar, Germany).

A primary antibody to MCMV was raised in rabbits and cross-adsorbed with MCMV-free Oh28 maize seed extract as previously described for maize fine streak virus [42]. Slides were dewaxed, rehydrated, blocked with 5% bovine serum albumin (BSA) in phosphate buffered saline (PBS) for 1 h and incubated overnight with the primary antibody (1:400 dilution in PBS, 0.2% BSA). After six washes with PBS, slides were incubated for 2 h at room temperature in Alexa Fluor™ 488 Goat-Anti-Rabbit (SFX Kit, Thermo Fisher Scientific, MA, USA) (1/600 dilution in PBS, 0.2% BSA), then washed six times with PBS and mounted with anti-fade media (Biomeda, GelMount, Foster City, CA, USA). Imaging was done on the Leica (Morrisville, NC, USA) DM IRB microscope equipped with a Leica DFC700T digital camera, or the Leica TCS-SP6 confocal microscope. MCMV infection levels in pericarp and pedicel tissue were estimated based on visual inspection of presence or absence of fluorescence using the scale: 1 = <5% tissue infection, few infection spots; 2 = 10–20% tissue infection; 3 = 20–50% tissue infection, MCMV abundant; or 4 = >50% tissue infection. Since MCMV was never detected in the endosperm or embryo of the whole seed sections, no scores were assigned. In total, 58 unique seed sections were examined from MCMV-positive seed lots and 17 from MCMV-negative seed lots.

## Evaluation of seedborne MCMV infectivity by vascular puncture inoculation

The infectivity of MCMV obtained from seed was evaluated using VPI based on methods described previously [43]. Briefly, mature MCMV susceptible Oh28 seeds were incubated in

water overnight at 22˚C. Seeds were embedded in Play-Doh (Hasbro, Inc. Pawtucket, RI, USA) and inoculated approximately 1 mm parallel to the embryo with 3 µl inoculum using an engraving tool. Seed were incubated on damp paper towels at 30˚C for 48 h and transplanted to 50-cell trays. Plants were tested for the presence of MCMV by ELISA 30 days post inoculation (DPI).

For seed produced from artificially inoculated plants in Hawaii, 20 remnant seed were soaked in 20 ml 1x Hanks' Balanced Salt Solution (HBSS, Gibco$^{TM}$). For the MCMV-HI seed lots, pedicel and pericarp tissues were isolated and homogenized in 40 ml HBSS and virions were further concentrated using the virion-associated nucleic acids method [43], leading to a final 200 µl virus suspension in HBSS. The solution was brought to 40 ml with HBSS, centrifuged to remove debris, and the supernatant was filtered (0.45 µm). The filtered samples were then centrifuged at 147,000 $g$ for 2.5 h at 4˚C. For seed obtained from artificially infected plants produced in Kenya, 20 seed were soaked overnight at room temperature in 10 ml 0.01 M K-phosphate, pH 7, then ground in a food processor. Samples were placed on ice for 1 h, then 3–5 ml subsamples were centrifuged at 5,000 $g$. The MCMV concentration of the supernatants were estimated [27] were used individually as inoculum for VPI. Two independent experiments using seed from Kenya (n = 25 inoculated seed per experiment) were performed. A lack of remnant seed prevented execution of a second experiment using seed from Hawaii (n = 50 inoculated seed).

Remnant seed (10 per line) from the seed lots used for immunocytochemical localization of MCMV were soaked overnight in 10 ml 0.01 M K-phosphate, pH 7, and the resulting solution was used for VPI and estimation of MCMV concentration as above. Two independent experiments (n = 25 inoculated seed per experiment) were conducted.

## Simulation of seedborne transmission with infectious MCMV inoculum

Oh28 seeds were soaked overnight at either 4˚C or 23˚C in MCMV-KS inoculum (2 seed/ml). Inoculum was prepared by grinding MCMV-KS infected leaf tissue at 21 DPI in 0.01 M phosphate buffer (1:5 w/v). Seed similarly soaked in 0.01 M K-phosphate buffer served as negative controls. Seeds were then planted in sterile soil in 50 or 288 cell flats and grown for 28 days as outlined above. Inoculum was tested for infectivity after incubation by rub-inoculation of Oh28 seedlings as outlined above. Experiments were replicated four times using 62–130 seed per replicate. Negative control seed (39–47) were planted into separate trays for each experimental replicate. At 28 DAP, plants were sampled, pooled, and tested for the presence of MCMV by DAS-ELISA as outlined above. Individual plants from MCMV positive pools were resampled and retested for MCMV by DAS-ELISA.

## Statistical analyses

When insufficient tissue was available for re-testing each individual plant in an MCMV positive pool, estimates of MCMV seed transmission ($p$) frequencies were calculated as:

$$p = 1 - (1 - Q)^{\frac{1}{n}} \times 100$$

where $Q$ is the proportion of positive pooled samples and $n$ is the number of plants in each pool [44]. Upper and lower bounds were calculated at a 95% confidence level based on the Wilson score interval for these positive seed lots [45].

The relative MCMV titer was determined among different seed tissues from mean sample absorbance in ELISA values, normalized by subtracting the mean absorbance of the corresponding negative controls, with a linear mixed-model analysis of variance (LMM) using the GLIMMIX function in SAS 9.4 (SAS, Inc, Cary, NC USA; Fig 1). The experiment number was

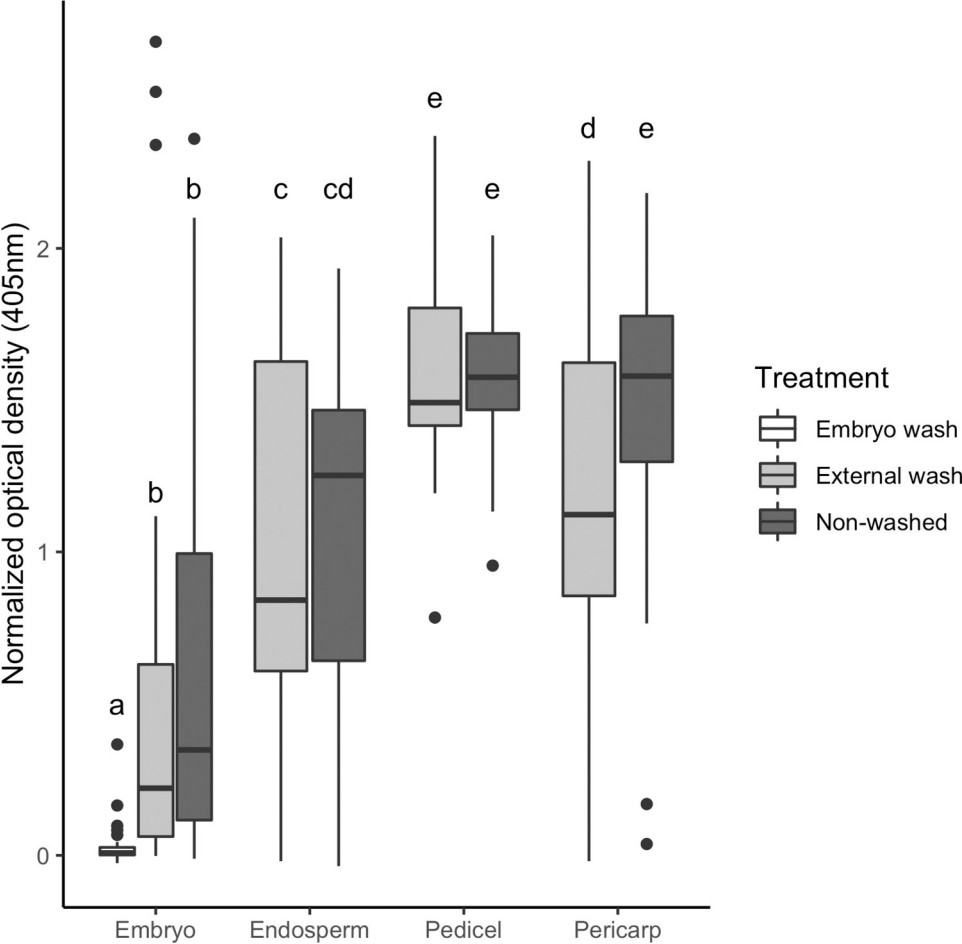

**Fig 1. Maize chlorotic mottle virus (MCMV) distribution in different dissected maize seed tissues.** Maize seed from MCMV infested seed lots were dissected into embryo, endosperm, pedicel, and pericarp tissues, then either processed directly for ELISA (Non-washed) or the whole kernel (External wash) or excised embryo (Embryo wash) was washed in 0.01% Triton X-100 containing 15% bleach for 15 min. prior to processing for ELISA. ELISA absorbance (405 nm) normalized to their respective negative controls are shown. Means for treatment*tissue with the same letter below the bar are not significantly different (p<0.05).

considered a random effect. Differences between treatments were assessed by the least squares method and visualize using ggplot2 in R 4.1.2 [46, 47].

## Results

### Transmission of MCMV through seed

Seed transmission of MCMV was tested using seed collected from MCMV-infected plants of maize inbred lines (CML333, CML442, CML545) from Hawaii and Kenya. All seed lots were verified MCMV positive by DAS-ELISA as described by Bernardo et al. (2021) [27]. Mean emergence rates ranged from 90–96% depending on seed lot (Table 1). From each lot, 94–98% of seedlings appeared healthy and of uniform size at time of sampling. Some seedlings did not develop well and were too small to sample individually (3,632 plants total). These were pooled for analyses within individual grow-out experiments.

In total, 85,163 seed collected from MCMV-inoculated plants from Hawaii and Kenya were grown and subsequently tested for the transmission of MCMV to developing leaves (Table 1).

**Table 1. Maize chlorotic mottle virus transmission from infested seed lots collected from Hawaii and Kenya.**

| Maize line | Origin | Plants tested | Moisture content | Emergence | Positive pools[a] | Transmission frequency | 95% confidence limits | |
|---|---|---|---|---|---|---|---|---|
| | | | | | | | Lower | Upper |
| | | | % | % | | % | % | % |
| CML333 | Hawaii | 21,001 | 16.3 | 90 | 0 | 0 | | |
| CML545 | Hawaii | 28,961 | 15.5 | 92 | 2[b] | 0.007 | 0.002 | 0.030 |
| CML333 | Kenya | 5,632 | 18 | 94 | 0 | 0 | | |
| CML442 | Kenya | 15,542 | 18.4 | 93 | 1[c] | 0.006 | 0.000 | 0.04 |
| CML545 | Kenya | 14,027 | 19.1 | 96 | 0 | 0 | | |
| Total | | 85,163 | | | 3 | 0.004 | 0.001 | 0.012 |

[a] Pools consisted of samples collected from up to 12 plants.

[b] Both pools identified as positive from an ELISA absorbance greater than twice the mean of healthy controls. The presence of MCMV was confirmed by RT-PCR, but insufficient tissue was available to test individual plants. Each pool consisted of samples collected from 10 plants.

[c] Identified as positive from an ELISA absorbance greater than the mean of healthy controls plus 3 times the standard deviation of the mean. The presence of MCMV in one plant among the 12-plant pool was confirmed by RT-PCR. The other plants were confirmed MCMV-negative.

No positive samples were identified from more than 26,000 CML333 seed collected from MCMV-inoculated plants from Hawaii and Kenya. Two pools of unhealthy seedlings (10 plants per pool) from CML545 seed obtained from inoculated plants from Hawaii were identified as positive for MCMV in ELISA (absorbance >2HC). The presence of MCMV-HI was confirmed by RT-PCR, sequencing, and reinfection of maize by rub inoculation. Based on the binomial model, the proportion of MCMV-HI infected CML545 plants was estimated at 0.007%. For CML545 seed lots from Kenya, two 10 plant seedling pools were positive for MCMV by DAS-ELISA (absorbance >2HC). However, in both cases only one of the two technical replicates was positive for MCMV, while the second was negative. The presence of MCMV was not detected by RT-PCR, suggesting the elevated absorbances were likely due to contamination or artifact. For CML442, the presence of MCMV in one pool of seedlings was identified as potentially positive from an ELISA absorbance >HC + 3SD. The presence of MCMV was confirmed by RT-PCR in a single plant from the 12-plant pool. Of thirteen other sample pools identified as positive for MCMV by an ELISA absorbance >HC + 3SD, none were positive when the pool was retested by RT-PCR and/or ELISA, and these are not included as positives in Table 1. Of the 85,163 plants tested, three pools were confirmed positive for the presence of MCMV, indicating an overall transmission frequency of 0.004%. An additional 4,296 seed collected from markets in Kenya (Bernardo et al 2021) and MCMV-inoculated plants in Hawaii were tested for transmission of the virus through seed using the same approach and none of the resulting plants were positive for MCMV.

## Detection of MCMV in dissected maize seed tissues

The presence of MCMV in hand-dissected samples of pericarp, pedicel, endosperm, and embryo tissues was evaluated by DAS-ELISA (Fig 1). Disinfestation of the seed surface by washing did not completely remove MCMV, however it significantly reduced detection in the pericarp (p<0.05). In non-washed samples, absorbances were significantly higher in the pedicel and pericarp compared to the endosperm and embryo (p<0.05; Fig 1). MCMV was not detected in embryos that were excised and washed (Fig 1; Table 2). These results indicate that, while MCMV is present in seed tissues, it does not appear to be evenly distributed and may be absent from the embryo.

**Table 2. Detection of maize chlorotic mottle virus (MCMV) in seed tissues by immunocytochemical microscopy.**

| Seed lot[a] | Origin | Positive seed sections[b] | Mean score[c] embryos | Mean score[c] endosperm | Mean score[c] pedicel | Mean score[c] pericarp |
|---|---|---|---|---|---|---|
| HI-A | Hawaii | 0/5 | 0 | 0 | 0 | 0 |
| HI-B | Hawaii | 0/5 | 0 | 0 | 0 | 0 |
| HI-H | Hawaii | 8/11 | 0 | 0 | 2.2 ± 1.5 | 2.3 ± 1.1 |
| HI-F | Hawaii | 5/10 | 0 | 0 | 2.8 ± 1.3 | 1.8 ± 0.8 |
| KE-C | Kenya | 0/7 | 0 | 0 | 0 | 0 |
| KE-D | Kenya | 10/13 | 0 | 0 | 1.1 ± 0.3 | 1.0 ± 0.0 |
| KE-F | Kenya | 8/11 | 0 | 0 | 2.0 ± 0.8 | 2.4 ± 0.7 |
| KE-G | Kenya | 13/13 | 0 | 0 | 3.2 ± 1.1 | 3.0 ± 1.0 |

[a]Seed lots from Hawaii and Kenya previously shown to be positive (HI-H, HI-F, KE-D, KE-F and KE-G) or negative (HI-A, HI-B, KE-C) for MCMV [27] were selected for further study.

[b]Number of positive sections/total sections.

[c]Longitudinal cross sections of seed were stained for the presence of MCMV by immunocytochemical microscopy as outlined in the Materials and Methods. Scores of fluorescence intensity were scored by visual inspection using the scale: 0 = 0% of tissue infected, 1 = less than 5% of tissue infected, few infection spots; 2 = 10 to 20% of tissue infected; 3 = 20 to 50% of tissue infected; 4 = more than 50% of the tissue is infected. Data shown are the mean ± S.D. MCMV was not detected in any endosperm or embryo sections.

## Immunolocalization of MCMV in seed

Although the dissection experiments suggested that MCMV was differentially distributed among seed tissues, it was not clear how much cross-contamination occurred during the dissection process. Therefore, immunocytochemical localization of the virus on seed tissues was carried out (Fig 2, Table 2). For sections from whole seed of five different MCMV positive seed lots from Hawaii and Kenya (HI-H, HI-F, KE-D, KE-F and KE-G; [27]), 44 of the 58 tissue sections that were evaluated had detectable immunofluorescence in the pericarp and pedicel (Fig 2) that was clearly different from the natural autofluorescence of the tissues. In contrast, sections of pericarp and pedicel from MCMV negative seed lots (HI-A, HI-B and KE-C) had low autofluorescence but did not have fluorescence associated with the immunofluorescent tag (Alexa fluor 488). Some variability in the abundance and intensity of fluorescence occurred within and among MCMV positive seed lots (Table 2). No fluor-specific fluorescence was detected in samples from the MCMV-negative seed lots HI-A, HI-B and KE-C. Similarly, none of the 59 embryos or 53 endosperm sections had detectable fluorescence. Fluorescence was observed in sections of pedicels and pericarps from MCMV positive seed lots HI-H, HI-F, KE-D, KE-F and KE-G. Although some variability was noted in the numbers of fluorescent sections scored from seed in these lots, a high degree of variability in scores was noted. These results confirm differential distribution of MCMV in seed being detected only in maternal pericarp and pedicel tissues in this study.

## Evaluation of seedborne MCMV infectivity by vascular puncture inoculation

Previous research indicated that MCMV could be detected in soak solutions and extracts of maize seed [27]. Here, the lack of MCMV transmission through seed found in the grow-out study was coupled with a clear detection of the virus in maternal tissues. To evaluate infectivity of MCMV from seed, MCMV-positive extracts and partially purified virus samples were inoculated into MCMV-free maize Oh28 seeds by VPI (Table 3). VPI transmits purified MCMV-KS virions to maize Oh28 seed with very high efficiency at concentrations as low as 1.00E-04 µg/µl virion protein and at lower efficiency for virion protein concentrations between

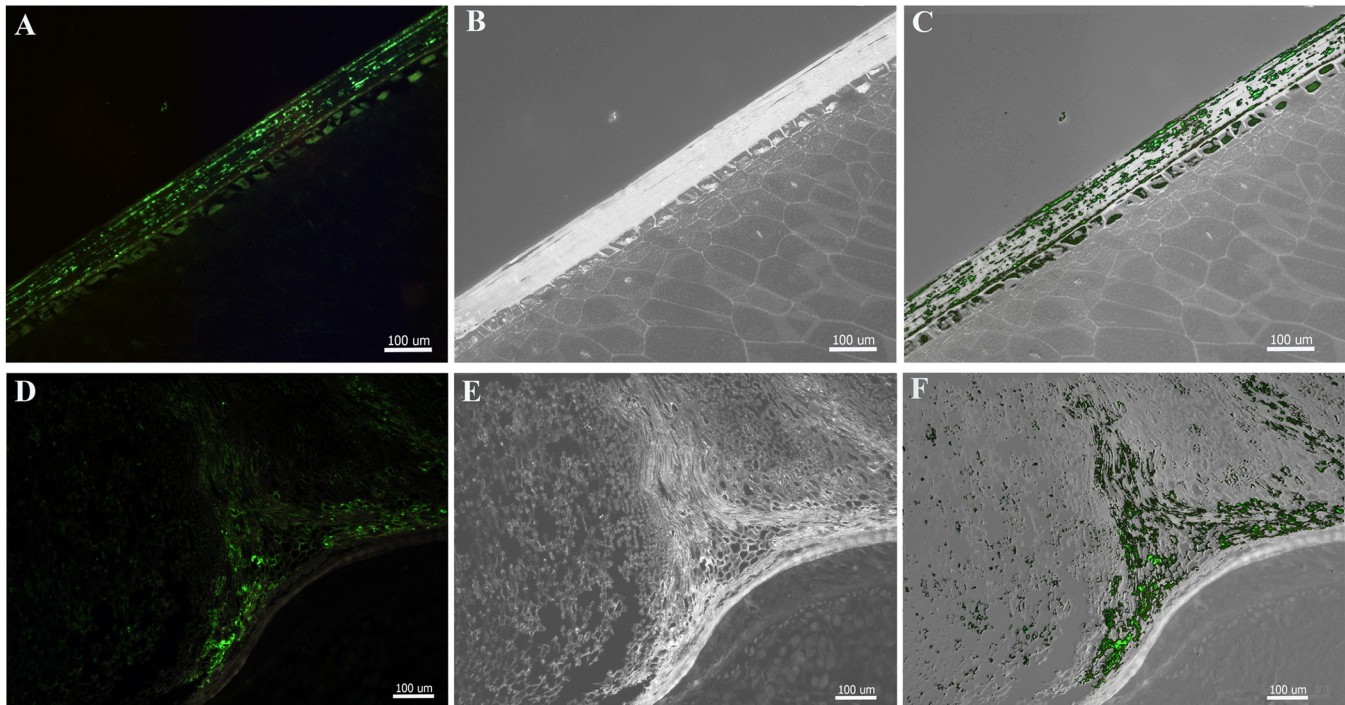

**Fig 2. Localization of maize chlorotic mottle virus (MCMV) in seed tissues sections.** Representative images of the distribution of MCMV in maize kernel pericarp (A—C) and pedicel (D—F) tissues as determined by fluorescent immunolocalizations using MCMV specific antibody followed by Alexa fluor 488 secondary antibody (green).

1.00E-05 and 1.00E-06 µg/µl (S2 Table). VPI of seed with extracts of MCMV-positive CML333 or CML545 seed used in the grow-out study, did not result in consequent infection of any of the resulting 146 seedlings. For CML442, 12 ± 7% of 51 inoculated seedlings were positive for MCMV by RT-PCR. For the seed lots used in localization experiments, samples from Hawaii seed lots were concentrated by ultracentrifugation prior to inoculation. Extracts of the three

**Table 3. Transmissibility of maize chlorotic mottle virus (MCMV) from seed by vascular puncture inoculation.**

| Inoculum Source | MCMV virion protein concentration | Emergence | Transmission frequency |
|---|---|---|---|
| | µg/µL | % | % |
| CML333-HI[a] | 6.6E-06 | 96 | 0 |
| CML545-HI[a] | 1.6E-04 | 96 | 0 |
| CML333-KE | 5.0E-04 ± 7.1E-04 | 61 ± 6 | 0 ± 0 |
| CML442-KE | 5.0E-04 ± 7.1E-04 | 73 ± 10 | 12 ± 7 |
| CML545-KE | 5.1E-05 ± 7.0E-05 | 77 ± 12 | 0 ± 0 |
| HI-H[a] | 1.1E+00 | 92 | 0 |
| HI-F[a] | 1.2E+00 | 88 | 0 |
| KE-D | 6.2E-01 ± 8.3E-01 | 100 ± 0 | 0 ± 0 |
| KE-F | 4.7E-02 ± 6.6E-02 | 100 ± 0 | 0 ± 0 |
| KE-G | 1.4E-03 ± 1.4E-03 | 94 ± 8 | 0 ± 0 |
| MCMV-KS[b] | 3.7E-02 ± 5.2E-02 | 88 ± 17 | 98 ± 2 |
| Phosphate buffer | 0.0 ± 0.0 | 70 ± 0 | 0 ± 0 |

[a]Insufficient seed to perform more than a single experiment.

[b]Detection of MCMV-KS by ELISA in positive control plants.

Kenyan lines were sufficiently concentrated for direct inoculation. No transmission of MCMV from these extracts to 192 inoculated maize seedlings was found. Positive and negative controls for each set of experiments behaved as expected: MCMV-KS leaf inoculum was transmitted successfully by VPI with 98 ± 2% efficiency and no transmission was observed for mock-inoculation with buffer.

In preliminary experiments, the impact of developmental stage and seed moisture on MCMV transmissibility via VPI was determined in seed harvested at three developmental stages (R2, R4, and R6) and three moisture levels in seed harvested at the R6 stage (S3 Table). For CML545 and CML333, inoculation with extracts from seed harvested at R2 resulted in 67 and 84% of inoculated seedlings testing positive for MCMV by RT-PCR, respectively. Similar inoculation with extracts of R4 seed produced 43% (n = 27/32) and 72% (n = 29/49) MCMV positive seedlings. The R6 seed was initially 23–24% moisture for both inbred lines. Inoculation of seed with extracts of CML333 R6 seed produced 41% (n = 16/39) MCMV positive seedlings, with the percentage of positive seedlings decreasing to 5% (n = 2/41) if the seed were dried to 17% moisture and 0% (n = 0/38) at 14% moisture. No MCMV positive seedlings were detected after seeds were inoculated with extracts of R6 CML545 seed, regardless of the moisture content (n = 189).

To test whether MCMV could be transmitted to germinating seedlings if infectious virus was present on the seed, MCMV-free seed (Oh28) was soaked in extracts from MCMV-KS inoculated seedling leaves overnight at 4˚C and 23˚C. Seeds were planted in greenhouse soil and moved to a growth chamber. At 28 DAP, plants were tested for the presence of MCMV by ELISA. Across four experiments for seed incubated in inoculum at 4˚C and 23˚C, 4.3 ± 2.1% (mean ± s.d.; n = 391) and 2.2 ± 2.9% (mean ± s.d.; n = 371) of the resulting seedlings were positive, respectively. None of 183 seedlings soaked overnight in extraction buffer alone were positive for MCMV by ELISA.

## Discussion

### Transmission of MCMV through seed

Serological tests conducted by DAS-ELISA on 85,163 seedlings derived from MCMV infected plants indicated the overall transmission frequency was 0.004%, demonstrating that MCMV is transmitted through seed at very low rates. Transmission was minimal among seed from plants inoculated with either the Hawaiian or the Kenyan MCMV isolate, indicating neither isolate is inherently more transmissible. This is not surprising given the limited diversity of MCMV isolates, which exhibit just 1–4% nucleotide divergence globally [12]. Seed transmission was not observed among CML333 seed at full maturity from either location, possibly due to the presence of MCMV tolerance in this line [48]. However, due to the low transmission frequency detected among all five seed lots, it is difficult to conclude that MCMV tolerant germplasm impacts seed transmission frequency.

Although we had originally planned to include temperate germplasm relevant to the U.S. corn belt in these transmission studies, these lines were unable to produce sufficient seed under tropical conditions and thus our experiments were limited to tropically adapted lines. Furthermore, we could not exclude possible potyvirus presence in field conditions where seed was collected for these experiments. The role of virus co-infection in seed transmission of MCMV and SCMV was not explored in this study and is poorly understood. However, MLN infected plants were recently found to not transmit MCMV or SCMV at rates higher than 0.017% [24]. Seed transmission of maize infecting potyviruses, SCMV and maize dwarf mosaic virus, is typically less than 1% [24, 49–51]. However, rates as high as 3.9–4.8% have been reported for SCMV in China [52, 53].

Transmission frequencies are consistent with previous studies conducted in the U.S. and abroad, which have reported that MCMV is seed transmissible at rates between 0 and 0.57% [20, 21, 24, 54]. Only three pools were confirmed positive by both DAS-ELISA and RT-PCR, indicating an overall transmission frequency of just 0.004% among the more than 85,000 seedlings tested (Table 1). Unlike many of the previously conducted grow-out tests, these experiments were performed under growth chamber conditions where potential vectors were actively excluded, which may explain the somewhat lower transmission frequency we detected compared with other studies. Only Uyemoto (1983) reported lower incidence of transmission, but fewer plants (4,051) were evaluated, which may have precluded detection. Higher MCMV transmission frequency was previously reported in Ecuador under growth chamber conditions at rates of 8–12% in two different maize hybrids. A diversity analyses of MCMV isolates collected globally found that Hawaii and Ecuadorian isolates were closely related and exhibited just 0.01 nucleotide diversity within this sub-group [11], suggesting that the Hawaii MCMV isolate used in this study may closely approximate strains present in Ecuador. It remains unclear how germplasm or seed handling may have influenced the higher transmission frequency reported by Quito-Avila et al. [7] Experimental conditions and sampling methods were not thoroughly described and no symptoms were observed among any virus positive seedlings [7].

## Localization of MCMV in seed tissue

Detection of MCMV among seed tissues by serological and immunolocalization tests revealed extensive viral presence among infested seed lots. Although MCMV was detected frequently in samples of maternal pedicel and pericarp tissues, it was not detected in endosperm or embryonic tissues evaluated among the five infested seed lots that had detectable levels of MCMV (Table 2, Fig 2). Despite the pervasive presence of MCMV amongst these seed lots, the localization of the virus to maternal tissues indicates it is much less likely to be transmitted through to subsequent seedlings given that it was not detected in the filial endosperm and embryo tissues. While the latter two tissues maintain living cells throughout seed maturation, where MCMV could theoretically retain infectivity, the pericarp and pedicel desiccate during maturation, leaving behind dead tissue that exists primarily for protection [55]. Thus, the lack of transmission of MCMV from seed tissues is likely related to its inability to replicate in these cells or move into surrounding tissues. Our findings are consistent with those that determined embryonic infection is generally needed to facilitate virus transmission via seed [28–30]. Interestingly, washing the seed in Triton/bleach solution significantly reduced the detection of MCMV in pericarp tissue by DAS-ELISA to levels not different from those found in the endosperm (Fig 1). These results suggest surface seed treatments may reduce MCMV contamination among seed and further exploration is needed to determine the efficacy of external seed treatments on disinfection of MCMV infested seed lots. Seed soaked directly in infectious MCMV inoculum and then planted immediately, indicated MCMV transmission rates were less than 5%. However, the long-term viability of this surface contamination and effects of drying require further exploration. While these low transmission rates suggest seed transmission of MCMV is unlikely to have a considerable impact on disease incidence in countries where the virus is already endemic, seed testing and field scouting are important tools that may prevent further spread of MCMV globally.

## Infectivity of seedborne MCMV

MCMV obtained from infested seed lots was found to be non-transmissible for nine of the ten lots tested using VPI, a procedure that is highly efficient for MCMV inoculation (Table 3).

These results suggest that despite detection of the virus using ELISA or PCR, the virus from seed is mostly non-infectious and/or non-transmissible. MCMV from a CML442 seed lot from Kenya was transmitted to six of 51 inoculated seedlings. This seed lot had a somewhat higher seed moisture content than desirable (18%), which may have contributed to the greater infectivity of MCMV.

Seed maturity stage and moisture content were found highly associated with MCMV transmissibility. Generally, the ability to transmit MCMV decreased using extracts of seed from ears of MCMV-inoculated plants harvested from R2 to R6 maturity and with decreasing moisture content in extracts of R6 seed (S3 Table). Because maternal tissues start to lose moisture during maturation, these results suggest that transmissibility of MCMV from seed is related to the moisture content and are unlikely to be explained by drying seed at 40˚C, since MCMV is remarkably stable and thermal inactivation is not achieved until 85˚C [56]. The impact of moisture on MCMV infectivity and severity was previously implicated as an important factor in its long-term retention in soil and plant residues and is associated with increased disease severity in wetter climates in East Africa [19, 24, 57]. These results suggest that full maturity and proper seed handling may be important for limiting transmission of MCMV through seed.

## Conclusions

Despite MCMV's low seed transmission, even a single infected plant can lead to disease outbreaks over time under favorable conditions [25, 26]. These factors include MCMV vector incidence, including maize thrips and chrysomelid beetles, as well as disease and vector reservoirs created by continuous maize plantings or alternative host presence such as sorghum or weed reservoirs, which act as "green bridges". These contributing factors are commonly present in tropical and sub-tropical climates where MLN epidemics have recently emerged. Given the large volume of seed traded globally, precautions are desirable for preventing the accidental introduction of MCMV to countries where MCMV is not endemic. Testing for the presence of MCMV using grow-out tests is both inefficient and costly and, given the low levels of seed transmission, unlikely to detect possible seed transmission. However, approaches for testing seed for the presence of MCMV using serological tests and molecular techniques that are highly sensitive have been developed [27] and allow for detection of seed that came from MCMV infected plants. Because these tests do not differentiate between viable virus in the embryo and inactivated particles in maternal tissue, they reflect that the seed likely came from a field containing MCMV-infected plants and may not be a good measure of relative risk for a given seed lot. In the absence of the ability to test seed in endemic regions, harvesting maize seed for replanting at full maturity and drying seed to less than 15% moisture content could be effective strategies for minimizing MCMV spread (S3 Table). Seed treatments including drying, chemical, heat, and storage conditions require further study to evaluate their impact on seed transmission of MCMV. Improved agronomic practices (e.g. roguing, harvest maturity, crop rotation) and seed treatments, together with the release and deployment of MCMV and potyvirus tolerant/resistant germplasm will allow for sustained and economically viable control of MLN.

## Supporting information

**S1 Table. Sensitivity of ELISA for detection of maize chlorotic mottle virus (MCMV).** (DOCX)

**S2 Table. Efficiency of maize chlorotic mottle virus (MCMV)-KS transmission by vascular puncture inoculation (VPI).** (DOCX)

**S3 Table. Transmissibility of maize chlorotic mottle virus (MCMV) from seed harvested from MCMV-HI infected plants at three maturity stages.**
(DOCX)

## Acknowledgments

The authors thank Mark W. Jones (USDA, ARS) for providing MCMV-free seed, Christian Mathias (Corteva Agriscience Wailaula, HI) for assistance with MCMV inoculation experiments, Kristen Willie (USDA, ARS) and Wilson Mwaura (CIMMYT) for their technical support, and Agdia Inc. (Elkhart, IN) for providing MCMV-HI infested seed.

**Disclaimer statement:** The findings and conclusions in this publication are those of the authors and should not be construed to represent any official USDA or U.S. Government determination or policy.

## Author Contributions

**Conceptualization:** Margaret G. Redinbaugh.

**Formal analysis:** Margaret G. Redinbaugh, Erik W. Ohlson.

**Funding acquisition:** Margaret G. Redinbaugh.

**Investigation:** Pauline Bernardo, Kelly Barriball, Timothy S. Frey, Tea Meulia, Anne Wangai, L. M. Suresh, Scott Heuchelin, Pierce A. Paul, Margaret G. Redinbaugh.

**Methodology:** Pauline Bernardo, Kelly Barriball, Pierce A. Paul, Margaret G. Redinbaugh.

**Supervision:** Pierce A. Paul, Margaret G. Redinbaugh.

**Visualization:** Tea Meulia, Erik W. Ohlson.

**Writing – original draft:** Margaret G. Redinbaugh, Erik W. Ohlson.

**Writing – review & editing:** Pauline Bernardo, Kelly Barriball, Timothy S. Frey, Tea Meulia, L. M. Suresh, Scott Heuchelin, Pierce A. Paul.

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
