## [Decision Letter · Decision Letter 0]

16 Nov 2022

PONE-D-22-27020Transmission, localization, and infectivity of seedborne Maize chlorotic mottle virusPLOS ONE

Dear Dr. Erik Ohlson,

Thank you for submitting your manuscript to PLOS ONE. After careful consideration, we feel that it has merit but does not fully meet PLOS ONE’s publication criteria as it currently stands. Therefore, we invite you to submit a revised version of the manuscript by 31 December 2022, that addresses the points raised during the review process.

 We look forward to receiving your revised manuscript.

Kind regards,

Islam Hamim, PhD

Academic Editor

PLOS ONE

Journal Requirements:

Additional Editor Comments:

Revision is required based on suggestions from the reviewers.

Reviewers' comments:

Reviewer's Responses to Questions

**Comments to the Author**

1. Is the manuscript technically sound, and do the data support the conclusions?

Reviewer #1: Partly

Reviewer #2: Yes

2. Has the statistical analysis been performed appropriately and rigorously? 

Reviewer #1: Yes

Reviewer #2: No

3. Have the authors made all data underlying the findings in their manuscript fully available?

Reviewer #1: Yes

Reviewer #2: Yes

4. Is the manuscript presented in an intelligible fashion and written in standard English?

Reviewer #1: Yes

Reviewer #2: Yes

5. Review Comments to the Author

Reviewer #1: The manuscript is a detailed study aiming to clarify the factors underlying the transmission process of MCMV, which along with infection by a potyvirus is a threat for maize production worldwide. The scientific soundness of the paper is not new, as most of the results confirm previous studies on the subject, but it clarifies some factors involved in the infectivity of seed-born MCMV.

I trust an important aspect involved in the seed transmission might be the virus strain and this should be more carefully detailed and discussed in the manuscript. How are considered the two MCMV strains used in this work? Are they mild or severe strains? This could be a factor influencing the contrasting seed transmission rates found between this work and the one by Quito Avila et al. 2016.

Despites quite well written, I will try to contribute by suggesting minor modifications.

Line 25. Follow ICTV rules for virus name citation; first letter should not be capitalized here.

Line 44. Virus species name must be italicized.

Lines 78-81. I trust this is Material and Methods.

Lines 107. Can light intensity be expressed in μmol?

Lines 108-109. Why different maize genotypes were used as positive and negative controls? They should carry the same genetic background to offer a uniform basis for evaluating symptoms.

Line 241. “none was”.

Line 252. Delete repeated “in no washed samples”.

Line 310. Suggestion: “for mock-inoculation with buffer”.

Line 318. Add “these” before “cells”.

Line 400. Delete “in this lot”.

Reviewer #2: This timely paper helps to clarify important gaps in understanding of seed transmission in MCMV with implications for other seedborne viruses. Quantifying seed transmission risk is complicated and fraught with pitfalls; I compliment the authors for the thoroughness of the work presented here and their thoughtful interpretations. I have only a few mostly minor comments, but one recommendation that is more important is related to the way the seed transmission frequencies are calculated, which I believe has led to overestimates.

Line 61 – The wording of this sentence is awkward; “crop virus epidemics” could be revised to “virus epidemics in crops.”

Line 140 – More details should be described about exactly how the tissue dissections were carried out. Some of this appears in the results section, but I think it should be described here. As the authors acknowledge, there is some risk of cross-contamination during this procedure, so it is important for the reader to understand the details of how it was performed.

Line 145 – Were embryos or other tissues pooled from multiple seeds before the ELISA testing? Or were individual embryos, etc., tested?

Lines 172-189 – This section lacks a description of the VPI procedure or even a reference to how it is performed. These are important details; a brief description and reference should be added.

Line 202 – This sentence is not informative without first understanding what Figure 1 presents. I suggest revising it to say, “The relative MCMV titer among different seed tissues was determined…(Fig. 1).”

Table 1 – “Emergence” would be more accurate than “germination.” Germination is a specific seed quality measurement that is measured under standard conditions, following established guidelines. The data shown here are for emergence under the conditions of the experiment.

Line 234 and Line 244 - Transmission frequency can be calculated and presented in a more accurate and precise way. It is possible to calculate a confidence interval around the estimates that are >0, and the ranges given for CML545 from Hawaii and the total of all seed lots are too high at the upper end. The upper bound of these estimates are based on the assumption that it would be possible that all seedlings in a positive pool might be positive (line 232). However, that possibility is exceedingly small, especially considering that 2,898 of the 2,900 pools (CML545) were negative. Assuming a binomial distribution of testing results (each pool can only be either positive or negative), with approximately 2,900 pools of 10 seeds each, if 2 pools are positive, the estimate of seed transmission frequency is indeed about 0.007% as reported by the authors. Tools are available to calculate confidence intervals around that estimate and by my calculations, the upper bound of the 95% CI for this estimate should be about 0.022%, NOT 0.07%. The probability that all twenty seedlings in both pools were infected (0.07% transmission frequency) is very low, about 0.00006%, or a 99.99994% CI. Similarly, the upper bound (95% CI) for the estimate of transmission frequency for CML442 from Kenya should be 0.031%, and for all 85,163 seeds, it should be 0.009%, not 0.02%.

Line 250 – Is “washing” the most appropriate term for this procedure? I would call it “disinfection” or “disinfestation.”

Line 285 – It would be preferable to include these results in Table 2 (even though they are all zeroes). That table will be used in the future to illustrate your results, and if there is no mention of embryo or endosperm tissues in the table, the take-home message could be lost.

Line 316 – Was VPI performed on mature seeds? A description of the procedure would clarify this.

Line 330 – These percentages are based on a total of how many seedlings?

Line 363 – It’s not clear what you mean by “conservative.” If you mean the estimates may be higher than actuality, I agree. Anyway, “conservative” could be interpreted by some readers as “low,” which is not what I think you mean.

Line 377 – There seems to be a word missing from this sentence.

6. PLOS authors have the option to publish the peer review history of their article (what does this mean?). If published, this will include your full peer review and any attached files.

Reviewer #1: No

Reviewer #2: **Yes: **Gary Munkvold

---

## [Author Response · Author response to Decision Letter 0]

3 Jan 2023

Reviewer and editor comments have been responded to in the response to reviewer document included in the resubmission. Thank you for consideration of our manuscript.

---

## [Decision Letter · Decision Letter 1]

25 Jan 2023

Transmission, localization, and infectivity of seedborne maize chlorotic mottle virus

PONE-D-22-27020R1

Dear Dr. Erik Ohlson,

We’re pleased to inform you that your manuscript has been judged scientifically suitable for publication and will be formally accepted for publication once it meets all outstanding technical requirements.

Kind regards,

Islam Hamim, PhD

Academic Editor

PLOS ONE

Additional Editor Comments (optional):

Reviewers' comments:

Reviewer's Responses to Questions

**Comments to the Author**

1. If the authors have adequately addressed your comments raised in a previous round of review and you feel that this manuscript is now acceptable for publication, you may indicate that here to bypass the “Comments to the Author” section, enter your conflict of interest statement in the “Confidential to Editor” section, and submit your "Accept" recommendation.

Reviewer #1: All comments have been addressed

2. Is the manuscript technically sound, and do the data support the conclusions?

Reviewer #1: Yes

3. Has the statistical analysis been performed appropriately and rigorously? 

Reviewer #1: Yes

4. Have the authors made all data underlying the findings in their manuscript fully available?

Reviewer #1: Yes

5. Is the manuscript presented in an intelligible fashion and written in standard English?

Reviewer #1: Yes

6. Review Comments to the Author

Reviewer #1: Authors have consistently answered the reviewers' comments and done the required changes. The revised version of the manuscript is now suitable for publication.

7. PLOS authors have the option to publish the peer review history of their article (what does this mean?). If published, this will include your full peer review and any attached files.

Reviewer #1: No

---

## [Editor Report · Acceptance letter]

27 Jan 2023

PONE-D-22-27020R1 

Transmission, localization, and infectivity of seedborne maize chlorotic mottle virus 

Dear Dr. Ohlson:

I'm pleased to inform you that your manuscript has been deemed suitable for publication in PLOS ONE. Congratulations! Your manuscript is now with our production department. 

Kind regards, 

on behalf of

Dr. Islam Hamim 

Academic Editor

PLOS ONE